# Zombie Agents: Detecting Semantic Livelock in Long-Horizon Autonomous Software

Simarjot Khanna
Independent Researcher
Canada
simarkhanna76@gmail.com

## Abstract

Long-horizon autonomous agents suffer from semantic livelock: they continue generating tokens and calling tools without making progress. Unlike a crash, this "zombie" state consumes API budget and time while remaining operationally active. We treat this as a progress violation and propose the Convergence Monitor, a lightweight sidecar that fingerprints agent states in embedding space. In a forensic analysis of real-world failures (SWE-agent corpus), we identified that 25% of the analyzed long-duration failures exhibited semantic livelock patterns. In one extreme case, an agent wasted 208 steps in a checkerboard oscillation pattern invisible to standard string-matching repetition guards. We argue that future Agentware requires a "liveness coprocessor" to ensure software makers do not pay for stalled execution.

## CCS Concepts

• **Computing methodologies** → *Artificial intelligence*; • **Software and its engineering** → **Software creation and management**.

## Keywords

autonomous agents, semantic livelock, liveness properties, runtime monitoring, AIware

**ACM Reference Format:**
Simarjot Khanna. 2026. Zombie Agents: Detecting Semantic Livelock in Long-Horizon Autonomous Software. In *Proceedings of the 3rd ACM International Conference on AI-Powered Software (AIware '26), July 6–7, 2026, Montreal, QC, Canada.* ACM, New York, NY, USA, 3 pages. https://doi.org/10.1145/3805760.3814895

## 1 Introduction

Foundation-model-powered agents [5] are increasingly being deployed as long-horizon autonomous software. Unlike traditional software with explicit control flow, agent execution is semantically driven and open-ended. An agent tasked with refactoring a module might attempt dozens of edit-test-undo cycles, accumulating API costs while producing no net change. No individual step triggers an error, yet the process has failed.

We term this failure mode **semantic livelock**: unlike deadlock [3], where execution halts, the agent remains operationally active but ceases to reduce its semantic distance to the goal. This is the "zombie" failure mode. The agent appears alive (generating novel

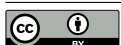

tokens and tool calls) but its semantic trajectory has flatlined. Existing frameworks rely on basic syntactic loop detection. Reflexion [6] and SWE-agent [7] use action-string matching or retry limits. These methods miss *paraphrased stagnation*, where an agent attempts semantically identical strategies using lexically distinct commands. We draw on the classical SE tradition of liveness properties [1, 2] to argue that agent reliability requires distinct monitoring for *progress*, separate from *safety* [8, 9].

**Contributions.** (1) A characterization of semantic livelock and three diagnostic patterns (Orbit, Oscillation, Diffusion); (2) The Convergence Monitor architecture; (3) A forensic analysis of real-world traces identifying semantic livelock in 25% of sampled long-duration failures; (4) A research roadmap for liveness-aware agents.

**Related Work.** Current frameworks primarily rely on syntactic loop detection [6, 7]. Beyond syntactic matching, several complementary paradigms exist. *LLM-as-a-judge* approaches use a secondary model or the agent's own reflection mechanism to evaluate whether progress is being made. However, these rely on the same class of reasoning capability that produced the stagnation. *Action-type sequence analysis* detects structural patterns (e.g., repeated edit-test-undo cycles) but misses paraphrased semantic stagnation, where the agent varies its commands while repeating equivalent strategies. *Process Reward Models* (PRMs) [14] provide step-level progress signals but require task-specific training data, unlike our task-agnostic, frozen-encoder approach.

## 2 The Convergence Monitor

We propose the Convergence Monitor as a lightweight sidecar to the agent loop (Figure 1). Following the runtime verification paradigm [4], it operates independently of the agent's internal reasoning, treating the agent as a black box that emits a stream of observations and actions.

**Liveness Formalization.** To anchor this architecture within classical liveness properties [1, 2], we define a progress predicate $P(\sigma, t)$ over agent trajectories: $P(\sigma, t)$ holds iff $\exists t' > t$ such that $\delta(e(t'), G) < \delta(e(t), G)$; that is, the agent eventually reduces its semantic distance to the goal. A liveness property requires $P(\sigma, t)$ to hold for all reachable states; semantic livelock is a violation of this property. Because most current agent architectures do not expose a goal embedding $G$, the Convergence Monitor serves as a runtime approximation of $P$, using rolling semantic diversity as a proxy.

### 2.1 State Fingerprinting and Diversity

At each step $t$, the monitor extracts a state embedding $e(t) \in \mathbb{R}^d$ from the agent's action-observation pair using a frozen encoder (e.g., `all-MiniLM-L6-v2`). Its low computational footprint allows the monitor to run on CPU in parallel with the agent's heavy GPU

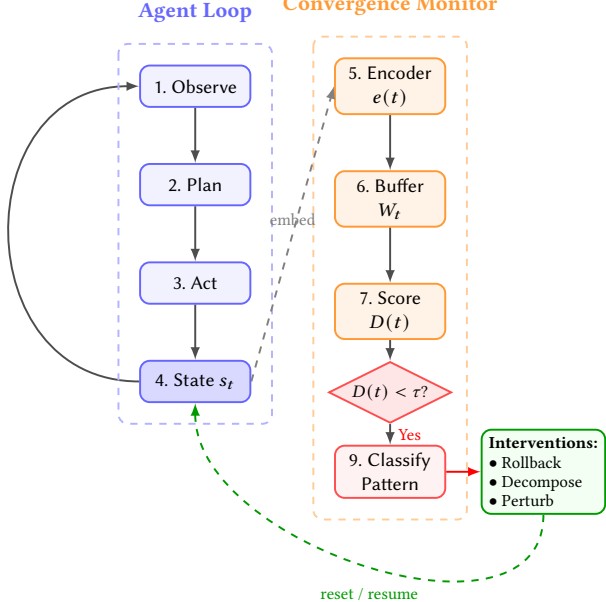

**Figure 1: The Convergence Monitor Architecture. State embeddings are extracted at each step. If diversity $D(t)$ drops below $\tau$, the Classifier triggers specific interventions.**

inference, adding negligible latency. We compute a rolling diversity score $D(t)$ over a window $W_t$ of size $w$:

$$D(t) = 1 - \text{mean}(\text{cosine\_similarity}(e(i), e(j))) \quad \forall i, j \in W_t, \ i < j \tag{1}$$

When $D(t) < \tau$ for $k$ consecutive steps, the monitor flags a potential livelock. Unlike syntactic de-duplication, this metric captures semantic redundancy even when the agent varies its wording.

## 2.2 Livelock Pattern Classification

Once triggered, the monitor classifies the stagnation pattern from the self-similarity matrix S:

- **Oscillation (The Checkerboard):** High similarity on off-diagonals, indicating the agent is toggling between two semantic attractors (e.g., $A \to B \to A$).
- **Orbit (The Block):** Dense blocks of high similarity along the diagonal ($S_{ij} \approx 1$), indicating a multi-step cycle through semantically equivalent states.
- **Diffusion (The Drift):** High mean similarity ($\mu > 0.8$) with no periodic structure, indicating the agent is wandering locally without directional progress. Note: Patterns that exhibit high similarity but do not match the structural criteria for Orbit or Oscillation are classified as Diffusion. This includes cases where the agent makes semantically redundant but lexically varied attempts.

## 3 Feasibility Study: Real Agent Failures

To demonstrate the viability of our approach, we applied the Convergence Monitor retrospectively to the *Nebius SWE-agent Trajectories* dataset [13], built on SWE-bench [12].

**Dataset Selection.** The SWE-bench automated evaluation harness defines failed trajectories: the agent's proposed patch did not resolve the target issue. From a streaming sample of the Nebius corpus [13], we identified 116 such failed trajectories, filtered to 60 candidates by requiring a minimum duration of 15 steps, and selected the top 20 longest (ranging from 38 to 155 steps) to maximize the probability of observing long-horizon stagnation.

**Prevalence & Patterns.** From the set of 20 longest failure trajectories, our monitor (configured with $w = 5, \tau = 0.15, k = 3$) flagged 5 cases of semantic livelock. While this 25% rate is specific to our sample of long-tail failures, these are precisely the most expensive errors in production. Systematic sensitivity analysis across these parameters remains future work.

- **Dominant Pattern:** We found that *Diffusion* (unstructured high-similarity drift) was more common than explicit Oscillation. Of the 5 flagged cases, 3 exhibited Diffusion-like patterns while 2 showed clear Oscillation.
- **Failure Hotspots:**
  A single repository instance (PyCQA__pyflakes-761) accounted for 45% of the top failed trajectories (9/20), suggesting that specific environment configurations can act as "livelock traps" for autonomous agents.

**Performance.** The monitoring overhead was negligible. Using the all-MiniLM-L6-v2 encoder, generating embeddings for a 155-step trajectory required approximately 13 seconds, confirming the sidecar's suitability for real-time deployment.

**Case Study: The Oscillating Agent.** We performed a deep-dive forensic analysis on a separate run of Instance TACC__agavepy-62, which ran for an extended 233 steps before failing.

- **Visual Evidence:** The Self-Similarity Matrix (Figure 2a) reveals a dense checkerboard structure. The mean pairwise similarity in the tail is **0.890**.
- **Semantic Freeze:** In even more extreme outliers (e.g., Melevir__cognitive_complexity-15), we observed semantic velocities ($v(t) = \|e(t) - e(t-1)\|$) as low as **0.034** with similarity **0.990**, effectively freezing the agent in state space.
- **Implication:** A deployed Convergence Monitor would have detected livelock onset at Step 25 (Figure 2b), marking the start of **208 steps** of wasted compute.

## 3.1 Threats to Validity

**Construct Validity.** We use low semantic diversity as a proxy for lack of progress. We acknowledge that high diversity does not guarantee progress (e.g., hallucination spirals). However, our manual inspection confirmed that sustained *low* diversity ($< 0.15$) consistently correlated with stalling.

**External Validity.** Our prevalence rate (25%) is derived from the Nebius SWE-agent corpus. Other agent architectures (e.g., ReAct vs. P-R) or base models may exhibit different livelock dynamics.

**Failure Mode Discrimination.** A successful trajectory may also exhibit decreasing semantic diversity as the agent converges on a correct approach. However, successful runs terminate rapidly once diversity drops, producing a valid patch and exiting. The distinguishing signature of semantic livelock is the *persistence* of

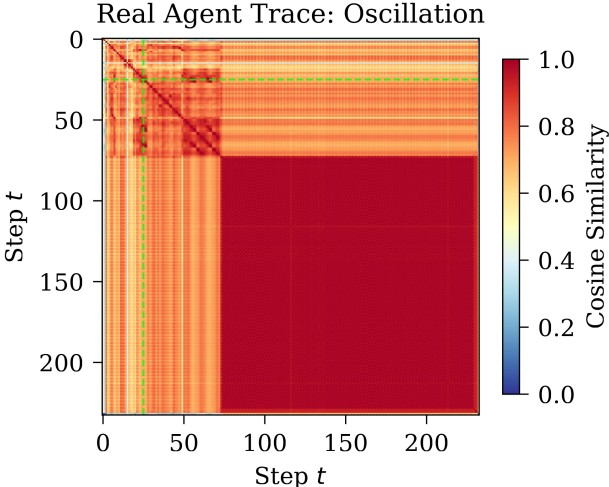

**(a) Self-Similarity Matrix (Oscillation)**

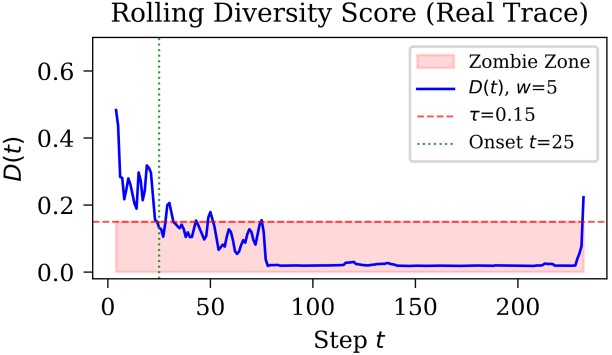

**(b) Diversity Score $D(t)$ Collapse**

**Figure 2: Forensic Analysis of TACC__agavepy-62. (a) The matrix reveals a dense "Checkerboard" pattern; the dashed lines mark the livelock onset. (b) The diversity score flatlines at Step 25, signaling the zombie state.**

low diversity without resolution. Comprehensive false-positive benchmarking against successful trajectory termination latencies remains important future work.

## 4 Vision: The Liveness-Aware Stack

**The Conscious Sidecar.** Current agents are "subconscious" text generators. We envision a standard architectural component, the Monitor, that acts as the agent's conscious awareness of time and progress. Future Agentware must standardize this interface so developers can plug in different "Liveness Policies" (e.g., strict for finance, loose for creative writing).

**Self-Healing Protocols.** While the interventions below remain a research agenda rather than evaluated contributions of this paper, upon detecting the *Orbit* pattern, the runtime should not just alert, but actively intervene. We propose a library of standard interventions: pruning the context window to remove the attractor state or injecting a "confusion" prompt to force exploration, inspired by curiosity-driven methods [11].

**Goal-Conditioned Progress.** Diversity is a proxy, not the goal. The next generation of monitors must incorporate the goal embedding $G$ directly into the progress metric, potentially using Bayesian change-point detection [10] for adaptive thresholding, to detect "hallucination spirals", where an agent generates diverse, novel, but completely irrelevant actions.

## 5 Conclusion

We have identified semantic livelock as an emerging failure mode in long-horizon agents. Through forensic analysis of real-world failures, we demonstrated that embedding-space trajectory analysis can reveal stagnation patterns missed by existing syntactic detectors. These findings suggest that "Liveness-Aware" agents could monitor their own semantic velocity and self-correct before wasting resources on zombie execution.

## Acknowledgments

We thank the anonymous reviewers of AIware 2026 for their constructive feedback. This work utilizes the Nebius SWE-agent Trajectories dataset. Generative AI tools (Large Language Models) were used to refine the text of this work, including grammar and clarity improvements. All conceptualization, data analysis, and scientific verification were performed by the authors.

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
