# OpenReview forum: "Zombie Agents: Detecting Semantic Livelock in Long-Horizon Autonomous Software"
_ACM.org/AIWare/2026/Conference — AIware 2026_

### Official Review · Reviewer_ZfzD · 2026-03-08

**Rating:** 3
**Confidence:** 4

**Review:**

## Pros
+ The proposed concept of semantic livelock is well-motivated. Framing it via classical liveness properties is conceptually elegant.
+ The lightweight sidecar design (frozen encoder, black-box monitoring) is practical and easy to integrate.
+ The three-pattern taxonomy (Oscillation, Orbit, Diffusion) is a useful vocabulary for characterizing agent stagnation.

## Cons
- Narrow related work discussion: only syntactic string-matching is considered as an alternative; other plausible detection strategies are not discussed.
- Shallow connection to classical liveness: the paper draws on liveness properties but does not formalize what "progress" means for agent execution.

## Comments

Thanks for submitting your paper to AIware 2026. The paper addresses a practically important problem: autonomous agents silently wasting API budget and compute in semantically stagnant loops. The concept of semantic livelock is well-motivated. The proposed sidecar architecture that uses embedding-space diversity as a lightweight, black-box monitoring signal is a practical and reasonable design. The three-pattern taxonomy (Oscillation, Orbit, Diffusion) provides a useful vocabulary for characterizing different stagnation behaviors. The forensic case study on TACC__agavepy-62 (208 wasted steps) is a convincing demonstration of the problem's severity.

However, I have two main concerns. First, the paper only compares against syntactic string-matching as the baseline detection approach (Reflexion's action-string matching and SWE-agent's retry limits), but does not discuss other plausible detection strategies, e.g., using the LLM itself as a progress judge (self-reflection) or action-type sequence analysis. A vision paper should help the reader understand the broader solution landscape instead of one specific mechanism. Second, although the paper anchors itself in classical SE liveness properties (citations [1] and [2]), the connection remains at the analogy level. The paper would benefit from formalizing what "progress" means in the context of agent execution. For a vision paper, this kind of conceptual deepening would be valuable.

**Summary:**

This paper identifies "semantic livelock" as a failure mode in long-horizon autonomous agents. In this mode, the agent remains operationally active, calling tools and consuming tokens, but it is unable to make meaningful progress. To detect such failure at runtime, the paper proposes Convergence Monitor, a lightweight sidecar that uses a frozen sentence encoder (MiniLM-L6-v2) to fingerprint agent states and compute a rolling diversity score. When diversity drops below a threshold, the monitor classifies the stagnation into one of three patterns: Oscillation, Orbit, and Diffusion. A retrospective forensic analysis of the 20 longest failure trajectories from the Nebius SWE-agent corpus revealed that five (25%) exhibited semantic livelock, with one case wasting 208 steps. The paper concludes with a vision for a "Liveness-Aware Stack" with pluggable liveness policies and self-healing protocols.

---

> ### Author Response · Authors · 2026-03-17
> **Thank you for the constructive feedback. Your two primary concerns highlight exactly where the paper can be most improved, and we will prioritize both in the revision.**
>
> Thank you for the constructive feedback. Your two primary concerns highlight exactly where the paper can be most improved, and we will prioritize both in the revision.
>
> On narrow related work:
> You are correct that we currently only compare against syntactic string-matching, such as Reflexion’s action-string matching and SWE-agent’s retry limits. We will expand the discussion to include a broader landscape of detection strategies. Specifically, we plan to add:
>
> - LLM-as-judge / self-reflection: While complementary, a key limitation is that the same reasoning capability that produced the stagnation is asked to diagnose it. Our sidecar operates on embeddings external to the agent's reasoning loop.
>
> - Action-type sequence analysis: Detecting structural patterns like edit-test-undo cycles. Useful when semantic repetition maps to repeated action types, but misses the paraphrased stagnation case (semantically identical strategies expressed through different action sequences), which is the specific gap our paper targets.
>
> - Process reward models: Which provide step-level reward signals that could flag non-progressing steps. Related problem, but PRMs require task-specific training data. Our approach is task-agnostic (frozen encoder, no training).
>
> We'll frame these as complementary strategies rather than competitors.
>
> On the shallow liveness connection:
> Your suggestion to formalize "progress" is well-taken and necessary. In the revision, we will:
>
> 1. Define a progress predicate $P(\sigma,t)$ that holds when the agent eventually reduces its semantic distance to the goal, i.e., $\exists t^{\prime}>t$ such that $\delta(e(t^{\prime}),G)<\delta(e(t),G)$.
>
>
> 2. State the liveness property as $P(\sigma,t)$ holding for all reachable states. Semantic livelock is the violation of this property: from some step $t$ onward, the agent never gets closer to $G$. The Convergence Monitor functions as a runtime monitor for an approximation of $P$, using diversity as a proxy because most current agent architectures do not expose a discrete goal embedding $G$. This level of formalization sets up a rigorous foundation for future work.
>
> We think this level of formalization fits a vision\new-idea paper and sets up the fully formal treatment as future work.

---

### Official Review · Reviewer_bx9M · 2026-03-10

**Rating:** 2
**Confidence:** 4

**Review:**

Positives
+ This paper addresses a practical issue in agents that may lead to unnecessary token usage.
+ The paper introduces a new pattern called semantic livelock and provides an initial experiment showing that it appears in some failed trajectories.

Negatives
- The dataset selection process is unclear.
- The empirical evaluation is very limited, as it analyzes only 20 trajectories.
- The experiment includes only failed trajectories and does not compare with normal (successful) trajectories.

Questions

1. In Abstract (line 15), the paper claims that “25% of long-duration failures were due to semantic livelock”. However, the feasibility study in Section 3 appears to only show that semantic livelock patterns are observed in some long-duration failure trajectories, rather than establishing that these failures were caused by semantic livelock. Could the authors clarify this statement?

2. The dataset selection in Section 3 (line 133) is unclear.
    - How is a trajectory determined to be a “failed trajectory”?
    - After identifying 116 failed trajectories, how are they filtered down to 60 candidates?
    - Why are only the top 20 longest trajectories (38–155 steps) selected for the analysis? Are there other longer trajectories that were excluded, and if so, why?
    - The dataset contains only 20 trajectories, which seems quite small.

3. The experimental dataset contains only failed trajectories. Why were normal (successful) trajectories not included in the analysis? The current experiment shows that semantic livelock appears in some failed trajectories, but it remains unclear how often semantic livelock occurs in normal trajectories. This comparison seems important, especially if semantic livelock is intended to be used as a proxy signal for problematic agent behavior.

Overall, the paper raises an interesting observation, but the current evaluation is too limited to demonstrate whether semantic livelock can serve as a proxy signal for problematic agent behavior.

**Summary:**

This paper introduces a pattern called semantic livelock in agents. The authors propose a Convergence Monitor on the agent loop to detect and classify such semantic livelock using embedding-based semantic diversity. An experiment on 20 failed trajectories reports that 5 of them exhibit semantic livelock patterns.

---

> ### Author Response · Authors · 2026-03-17
> **We appreciate your push for empirical rigor. We will ensure the final text carefully frames the Convergence Monitor as identifying a specific, wasteful failure pattern, while explicitly deferring the comprehensive empirical validation (control groups, cross-architecture tests, and false-positive rates) to future work, as is appropriate for the scope of a vision paper.**
>
> Thank you for the detailed feedback and questions. We appreciate the opportunity to clarify our methodology and will ensure these points are explicitly addressed in the revision.
>
> On the causal claim (Q1): You're right that "were due to" overstates what we showed. We'll revise to
> "exhibited." Importantly, even without establishing causality, detecting this state remains highly valuable for cost-saving. For example, in our TACC_agavepy-62 case study, the agent entered an A-B-A oscillation at Step 25 and wasted a subsequent 208 steps (mean pairwise similarity 0.890). A monitor catching this pattern saves compute regardless of whether the live-lock was the root cause or a downstream symptom of the failure.
>
> On dataset selection (Q2): We apologize for the lack of clarity in Section 3 regarding our data sampling and will fix this in revision. To clarify your questions:
>
> - "Failed" trajectories are strictly defined by the SWE-bench automated evaluation harness (where the agent's proposed patch did not resolve the issue), rather than by manual judgment.
>
> - Our filtering process started by identifying 116 failed trajectories from the streaming sample, which we then narrowed to 60 candidates by requiring a minimum duration of 15 steps.
>
> - Why only the 20 longest? These are the 20 longest trajectories in the candidate pool (ranging from 38 to 155
> steps). Nothing longer was excluded, this is the full tail. We targeted the longest failures because semantic live-lock is a long-horizon phenomenon, and short failures terminate before it can manifest.
>
> - We acknowledge that a sample size of 20 is limited for population-level claims. Because this was submitted to the short vision/new ideas track (2-4 pages), our goal with Section 3 was specifically a feasibility demonstration to show these patterns can be detected in real traces, rather than a definitive prevalence study.The separate deep-dive on TACC_agavepy-62 (233 steps) provides additional qualitative depth.
>
> On the lack of successful trajectory comparison (Q3): Fair point. We'd expect a successful trajectory to show
> decreasing diversity as the agent converges on a correct approach, followed by rapid termination (the agent produces a valid patch and exits). The distinguishing feature of live-lock is that low diversity persists for many steps without resolution: the agent keeps going but never produces a solution. In the TACC case, this gap was 208 steps. A full false-positive analysis on successful trajectories is the natural next step, and we'll add discussion of the expected behavior and this termination-latency distinction in revision.
>
> Overall:
> We appreciate your push for empirical rigor. We will ensure the final text carefully frames the Convergence Monitor as identifying a specific, wasteful failure pattern, while explicitly deferring the comprehensive empirical validation (control groups, cross-architecture tests, and false-positive rates) to future work, as is appropriate for the scope of a vision paper.

---

> > ### Comment · Reviewer_bx9M · 2026-03-20
> >
> > Thank you for the detailed clarifications.
> > The responses help improve the clarity of the dataset construction and the intended scope of the paper.
> >
> > However, I still have concerns regarding the limited empirical evaluation (e.g., small sample size and lack of comparison with successful trajectories).

---

### Official Review · Reviewer_YJ1t · 2026-03-11

**Rating:** 4
**Confidence:** 4

**Review:**

# Pros

- The notion of semantic livelock is well-motivated and resonates strongly with real-world experiences of deploying long-horizon agents.
- The Convergence Monitor is designed as a sidecar that treats the agent as a black box, requiring no access to internal reasoning or model states. The use of frozen, lightweight sentence embeddings and CPU-side computation makes the approach practical and easy to integrate into existing agent frameworks.
- The problem is real and non-trivial.
- The proposed categorization of livelock patterns (e.g., oscillation, orbit, diffusion) provides a helpful lens for reasoning about agent failures and could facilitate future work on diagnosis, benchmarking, and recovery strategies.

# Cons
- The core assumption that decreasing semantic diversity in embedding space correlates with lack of task progress is intuitive but not rigorously justified.

- The choice of window size, diversity threshold, and consecutive-step criteria appears ad hoc. The paper does not provide sensitivity analysis or guidance on how these parameters generalize across tasks, domains, or agent architectures.

- While the paper sketches possible interventions (rollback, decomposition, perturbation), these are not evaluated empirically. As a result, the work stops short of demonstrating that detecting semantic livelock leads to improved agent outcomes.

**Summary:**

This paper identifies and formalizes a novel failure mode in long-horizon autonomous agents, termed semantic livelock, where an agent remains operationally active (continuing to generate tokens and invoke tools) but makes no semantic progress toward the task objective. The authors propose a runtime monitoring component, the Convergence Monitor, which detects semantic stagnation by tracking diversity in embedding space over a rolling window.
Through a forensic analysis of failure traces from the SWE-agent corpus, the paper reports that a non-trivial fraction of long-horizon failures exhibit semantic livelock patterns that are not captured by existing syntactic loop-detection heuristics. The paper further categorizes semantic livelock into several recurring patterns (e.g., oscillation, orbit, diffusion) and discusses potential intervention strategies.